# Root Na^+^ Content Negatively Correlated to Salt Tolerance Determines the Salt Tolerance of *Brassica napus* L. Inbred Seedlings

**DOI:** 10.3390/plants11070906

**Published:** 2022-03-29

**Authors:** Cheng-Feng Wang, Guo-Liang Han, Zi-Qi Qiao, Yu-Xia Li, Zong-Ran Yang, Bao-Shan Wang

**Affiliations:** Shandong Provincial Key Laboratory of Plant Stress, College of Life Sciences, Shandong Normal University, Ji’nan 250014, China; wywcf123@163.com (C.-F.W.); adg129@126.com (Z.-Q.Q.); liyx202103@163.com (Y.-X.L.); yangzongran@126.com (Z.-R.Y.)

**Keywords:** *Brassica napus* L., correlative analysis, growth, ion content, salt tolerance

## Abstract

Soil salinization is a major environmental stressor that reduces the growth and yield of crops. Maintaining the balance of ions under salinity is vital for plant salt tolerance; however, little is known about the correlation between the salt tolerance of crops and the ion contents of their roots and shoots. Here, we investigated the poorly understood salt-tolerance mechanisms, particularly regarding ion contents (particularly Na^+^), in *Brassica napus* subsp. *napus* L., an agriculturally important species. Twenty *B. napus* inbred lines were randomly chosen from five salt-tolerance categories and treated with increasing concentrations of NaCl (0–200 mmol) for this work. We found that the root Na^+^ content is the most correlated limiting factor for the salt tolerance of *B. napus*; the higher the salt tolerance, the lower the root Na^+^ content. Correspondingly, the Ca^2+^/Na^+^ and K^+^/Na^+^ ratios of the roots were highly correlated with *B. napus* salt tolerance, indicating that the selective absorption ability of these ions by the roots and their translocation to the shoots play a pivotal role in this trait. These data provide a foundation for the further study of the molecular mechanisms underlying salt tolerance and for breeding salt-tolerant *B. napus* cultivars.

## 1. Introduction

The problem of soil salinization is long-established, dating from well before the evolution of humans and the development of agriculture, but it has been exacerbated by human agricultural practices such as poor irrigation, excessive fertilization, plowing, and climate change [1,2,3,4,5]. The area of saline–alkali land (including various saline and alkaline soils) worldwide is increasing by a rate of about 1 × 10^6^ to 1.5 × 10^6^ hm^2^ per year [6], already covering over 900 million hectares [7,8] or approximately 10% of the total global land area. It is estimated that by 2050, more than 50% of arable land will face salinization problems [9,10]. The average of saline soil in China is significantly higher than the global average, with about 3.6 × 10^7^ hm^2^ of salinized soil, and 6.62% of the country’s arable land is affected by salinization [11,12,13]. Saline soils (particularly Na^+^ in there) are often unusable for agriculture [14,15], meaning the amount of land available for growing crops is decreasing at the same time as the global population is rapidly growing to a predicted 9 billion people by 2050 [16]. This poses the major challenge of meeting food requirements around the world; therefore, it is very important that we learn how to utilize and improve saline soils to increase crop production.

For plants, high salinity leads to two direct stresses: ionic and osmotic stress. Ionic stress disturbs the ionic homeostasis of the cells, altering the plant’s physiological metabolism and thereby inhibiting normal growth and development [17,18,19], even leading to plant death. Excess Na^+^ is a major problem for plant cells as it induces a series of secondary stresses, such as deficiencies in nutrient minerals (particularly Ca^2+^ and K^+^) and oxidative stress [7]. Osmotic stress (the lowering of soil water potential) limits water uptake of plant cells which triggers a series of deleterious responses such as growth reduction, stomata closure, germination stagnation, and oxidative damage [20]. To survive in saline environment, plants have evolved different strategies such as salt exclusion, ion compartmentalization into vacuoles, and osmotic adjustment [20]. Recent work indicates that plant cells can sense Na^+^ and regulate ion transport by the combination of glycosyl inositol phoshorylceramide (GIPC), Ca^2+^ signaling, and Na^+^ transport [21]. The efficiency of these regulatory pathways determines ion content of roots and shoots, ultimately determines the salt tolerance of plants.

It is vital to elucidate salt tolerance mechanisms for breeding reliable salt-tolerant crop cultivars that can grow on saline soils [22]. Salt tolerance and its underlying mechanisms vary extensively depending on plant species, growth, and developmental stages, salt compounds, and concentrations, and environmental factors. Many studies have examined the indicators of salt tolerance related to crop growth. Crops with a strong salt tolerance tend to have longer roots and greater fresh weights than those with weak salt tolerance [23,24]. Under salt stress, sustained crop growth and yields are generally considered the main evaluation criteria for determining crop salt tolerance and developing salt-tolerant cultivars [25,26,27]. In addition to growth and physiological indicators, salt-tolerance traits may be associated with the ion content of plants grown in the presence of salt [28]. There have been numerous reports detailing the balance of ions (Na^+^, K^+^, and Ca^2+^, among others) related to plant salt tolerance [15,29,30]. In bread wheat, shoot ion-independent tolerance is considered an important trait for salt tolerance, favoring tissue expansion and tillering of plants before the shoots are affected by salt toxicity [31,32]. In general, the Na^+^ content in roots and shoots of a non-halophyte will increase to varying degrees under salt stress, while the contents of K^+^ and Ca^2+^ will decrease to varying degrees [33,34,35,36]. These ionic parameters are often used as indicators of plant salt tolerance. However, to date, there is no unified conclusion about the correlation between ion content and crop salt tolerance, particularly in the roots and shoots of crop plants.

*Brassica napus* subsp. *napus* L. (rapeseed) is one of the most important oil crops in the world. Its seed oil has high nutritional value, being rich in fatty acids and vitamins, and is also widely used in industry [37,38,39,40]. *B. napus* is generally considered moderately salt tolerant, with different varieties showing substantial differences in salt tolerance [41]. Different indicators, such as growth, water content, membrane injury index, enzymatic activities, and ion contents, have been used to assess the salt tolerance of different *B. napus* varieties. The relative salt stress–induced reduction in the biomass of sensitive varieties was found to be significantly higher than that of the tolerant varieties [42], while the K^+^ content and K^+^–Na^+^ selectivity were also important for salt tolerance [43]. Some reports suggest that plant salt tolerance depends mainly on the ability of the root system to limit the transport of Na^+^ to the shoots [43,44,45].

Relatively few studies of salt tolerance have collected large numbers of samples for testing, despite the expectation that the salt tolerance of a particular species or different varieties could be evaluated more efficiently through the use of a larger amount of sample data. Recently, we evaluated the salt tolerance of 549 *B. napus* inbred lines and identified five categories of salt tolerance [46]. In the current work, a total of 20 *B. napus* inbred lines were randomly chosen from the five salt-tolerance categories, which were studied to reveal the underlying mechanisms with an emphasis on the correlation between ion content in the root and shoot and salt tolerance under salt (NaCl) stress. Our results indicate that root Na^+^ content, highly negatively associated with salt tolerance, is the most reliable indicator for assessing salt tolerance in *B. napus* inbred seedlings.

## 2. Materials and Methods

### 2.1. Plant Materials

The seeds used in this experiment were harvested in May 2018 and stored at 6 °C.

Four *B. napus* inbred lines were randomly selected from each of the five salt-tolerance categories [46] (high salt sensitivity, HSS; salt sensitivity, SS; moderate salt tolerance, MST; salt tolerance, ST; and high salt tolerance, HST) as biological replicates, for a total of 20 inbred lines. The following numbers of the seeds were used in our work. HSS: 557, 611, 668, 703; SS: 40, 74, 277, 323; MST: 39, 319, 388, 525; ST: 130, 344, 527, 667; HST: 137, 370, 460, 475.

### 2.2. Growth Conditions

Healthy and uniform *B. napus* seeds were presoaked in 8 mL of deionized water for 10 h, then sown on two layers of filter paper in 10 cm Petri dishes. In the morning of the first two days, the water in the dish was replaced with a fresh 8 mL of deionized water. From the third day onwards, the NaCl concentration was gradually increased by replacing the water with a salt solution every 12 h, increasing by 50 mmol (mM) each time to a final concentration of 50, 100, 150, or 200 mmol (mM). The NaCl was dissolved in 1/5 Hoagland solution (the pH of all solutions was adjusted to 6.0). Seeds cultured in 1/5 Hoagland solution were the control.

The seeds were cultured in a phytotron with a temperature of 28 ± 3/23 ± 3 °C (day/night), a light intensity of 600 μmol m^−2^ s^−1^ (14 h light/10 h dark), and a relative humidity of 70% [46]. Three biological replicates were performed.

### 2.3. Determination of the Growth Indicators

The following growth indicators were measured on the 9th day after sowing (Appendix A). Shoot length (SL) and root length (RL): The aboveground part and the underground part (rhizome) of the seedlings were measured separately. After the leaves and roots were straightened out, a ruler was used to measure the lengths of 10 individuals for each biological repetition. Fresh weight of shoots (SFW), fresh weight of roots (RFW), and total fresh weight (TFW): The seedlings were rinsed with deionized water, wiped quickly with absorbent paper, divided into the shoots and roots, and weighed. TFW was sum of the SFW and RFW. Dry weight of shoots (SDW), dry weight of roots (RDW), and total dry weight (TDW): The shoots and roots were fixed at 105 °C for 15 min, dried at 80 °C until their weight was stable, then weighed separately. TDW was the sum of SDW and RDW. Dry weight to fresh weight of shoots ratio (S-DW/FW) and dry weight to fresh weight of roots ratio (R-DW/FW): These ratios were calculated as the dry weight over the fresh weight in the respective tissues. Based on these data, various relative values were also obtained (Appendix A).

### 2.4. Na^+^, K^+^, and Ca^2+^ Analysis

The Na^+^, K^+^, and Ca^2+^ contents were measured according to Wang and Zhao [11], with some modifications. Each sample of dried fine-ground roots or shoots was incubated with 1 mL nitric acid, extracted in a boiling water bath for about 3 h, filtered, and diluted to 10 mL using ultrapure water. The ion contents were measured using an AP1500 flame photometer (Shanghai Aopu analytical instruments co., Ltd., Shanghai, China).

The selective ion transport capacity was analyzed as described previously [47,48]. The followings were measured and analyzed (Appendix A): Na^+^, K^+^, and Ca^2+^ contents (mmol·g^−1^_DW_); K^+^/Na^+^ and Ca^2+^/Na^+^ ratios; ratio of the Na^+^ content in the shoot to that in the root (S_Na_); ratio of the K^+^ content in the shoot to that in the root (S_K_); ratio of the Ca^2+^ content in the shoot to that in the root (S_Ca_); and the ratios of the K^+^/Na^+^ ratio in the shoot to that in the root (S_K, Na_) and the Ca^2+^/Na^+^ ratio in the shoot to that in the root (S_Ca, Na_).

### 2.5. Statistical Analysis

The experiments performed here focused on the correlative analysis between growth parameters of root and shoot and their Na^+^, K^+^, Ca^2+^ parameters. Five salt concentrations (0, 50, 100, 150, and 200 mmol NaCl) and five salt tolerance levels (HSS, SS, MST, ST, and HST are treated as 1, 2, 3, 4, and 5, respectively) were quantified to construct a mathematical model for linear regression analysis. The correlation coefficients (*R*^2^) between the salt tolerances and ion contents of the roots and shoots of the *B. napus* lines belonging to the five salt-tolerance categories were analyzed from three aspects: (1) the correlation between the NaCl concentration and the individual ion content parameters in the roots and shoots of a particular salt-tolerance grade of *B. napus* inbred line (the *y* coordinate is the ion content parameter, and the *x* coordinate is the NaCl concentration); (2) the correlation between the salt-tolerance grade of *B. napus* inbred line and the individual ion content parameters in the roots and shoots of seedlings grown at a given NaCl concentration (the *y* coordinate is the ion content parameter, and the *x* coordinate is the salt-tolerance grade); (3) the correlation between growth index and the ion content parameters of all *B. napus* varieties under all treatments (the *y* coordinate is the ion content parameter, and the *x* coordinate is the growth indicator value).

Statistical and plotting analyses were performed using standard tools, such as Microsoft Excel 2010, GraphPad Prism 6.01, and SPSS 19.0 [49,50]. Specific mathematical tools were used, including the One-Way ANOVA and linear regression analysis. In the figures, different lowercase letters represent significant differences among the means (at a *p* < 0.05 level), as determined using Duncan’s test. In the tables, asterisks (*, **) represent significant differences among the means at a *p* < 0.05 level or 0.01 level, respectively.

## 3. Results

### 3.1. Growth under Varying Degrees of NaCl Stress

Growth is one of the most reliable indicators for assessing plant salt tolerance. Various growth indicators in *B. napus* seedlings grown under varying degrees of NaCl stress were determined (Appendix A). In general, the growth parameters of all inbred lines decreased with increasing NaCl concentrations, with the reduction of growth in the salt-sensitive inbred lines being greater than that of the salt-tolerant ones. The phenotypes of the *B. napus* seedlings were determined under different NaCl concentrations and salt-tolerance grades (Figure 1a). In particular, the total fresh weight was analyzed (Figure 1b), because this characteristic was previously shown to be the most correlated with *B. napus* salt tolerance [46]. No significant differences in phenotypes such as the total fresh weight were observed between the control and 50 mmol NaCl-treated seedlings; however, at 100 mmol NaCl, a small amount of necrosis was detected in the leaves. At 150 mmol NaCl, seedling growth was significantly damaged, while at 200 mmol NaCl, the damage was more serious and the leaves showed signs of wilting and yellowing, with salt-sensitive inbred lines showing more injury symptoms than salt-tolerant ones (Figure 1a). In all five salt-tolerant grades, the total fresh weight of seedlings decreased as the NaCl concentration increased, but for the inbred lines with higher salt tolerance, the total fresh weight of seedlings was significantly higher than those with lower salt tolerance (Figure 1b).

### 3.2. Correlation between the Relative Values of the Growth Indicators and NaCl Concentrations

Under NaCl stress, the relative value (treatment/control) represents the tolerance of a given genotype of a plant species: the higher the relative value, the higher the salt tolerance. We therefore also calculated the relative values of plant growth indicators (Appendix A) and performed a correlation analysis between the NaCl concentration of the growth medium and the relative value of each indicator (Figure 2). Our linear analysis indicated that both the relative values of SDW/SFW and RDW/RFW were positively correlated with the NaCl concentration (Figure 2i,j), whereas those of the other indicators were negatively correlated with the salt concentration (Figure 2a–h). In general, the *B. napus* inbred lines with lower salt tolerances showed greater changes in the relative values than those the lines with higher salt tolerances under different salt concentrations.

### 3.3. Correlation between the Na^+^, Ca^2+^, and K^+^ Content Parameters and the B. napus Salt Tolerance

Within each salt-tolerance grade, we found that the ion contents, absorption, and transport in the roots and shoots of the different varieties were related to the NaCl concentration in the growth medium. Among the five different salt-tolerance grades of the *B. napus* lines, those with moderate salt tolerance showed the strongest correlation between the ion content parameters and the NaCl concentration of the growth medium. The highest correlation coefficients were those of the shoot Na^+^ content with the NaCl concentration (*R*^2^ = 0.95 for moderately salt-tolerant varieties, *R*^2^ = 0.94 for highly salt-sensitive and highly salt-tolerant varieties), followed by those of the root Ca^2+^ content with the NaCl concentration (about 0.9~0.95), while the lowest correlation coefficients were those between S_K, Na_ and the NaCl concentration (Table 1). There was no significant regularity in the *R*^2^ values between the ion content parameters in different salt-tolerance grades of *B. napus* and the NaCl concentration of the growth medium. Together, the shoot Na^+^ content was highly positively correlated with the NaCl concentration.

Correlation between ion parameters of the *B. napus* varieties and their salt tolerances was shown in Table 2. Among the NaCl concentrations, the 200 mmol NaCl stress showed the highest correlation with the ion contents. Among the ion content parameters, root Ca^2+^/Na^+^ showed the highest correlation with the salt tolerance of *B. napus* varieties. The highest correlation coefficients were between root Ca^2+^/Na^+^ and the salt-tolerance grade of *B. napus* inbred lines (*R*^2^ = 0.78 for grown under 150 mmol NaCl stress, *R*^2^ = 0.77 for grown under 100 mmol NaCl stress), followed by those between the root Na^+^ content and the salt-tolerance grade of the *B. napus* inbred lines (*R*^2^ = 0.75 for grown under 200 mmol NaCl stress, *R*^2^ = 0.74 for grown under 150 mmol NaCl stress). The lowest correlation coefficients were those between the S_Ca_ and the salt-tolerance grade of *B. napus* inbred lines grown under NaCl stress (Table 2).

We also found that the ion contents were related to the growth indexes of *B. napus* inbred lines grown under different levels of NaCl stress (Table 3). In general, among the growth indexes, root length showed the highest correlation with the ion contents. Among the ion content parameters, the root Na^+^ content showed the highest negative correlation with the growth indexes. The highest correlation coefficient was found between root Ca^2+^/Na^+^ and root length (*R*^2^ = 0.78), followed by that between root Ca^2+^/Na^+^ and root fresh weight (*R*^2^ = 0.77), while the lowest correlation coefficients were those between S_K, Na_ and the growth indexes (Table 3). These results indicated that root growth including the fresh weight and elongation of *B. napus* inbred lines is highly positively correlated with the root Ca^2+^/Na^+^ ratio under NaCl stress, which is a reliable indicator for evaluating the *B. napus* salt tolerance under salt stress.

The Na^+^, Ca^2+^, and K^+^ contents in the shoots and roots of the plants reflected their ion uptake, translocation, and compartmentalization, as well as their salt tolerance. The Na^+^, K^+^, and Ca^2+^ contents; K^+^/Na^+^ and Ca^2+^/Na^+^ ratio; ratios of the Na^+^ content in the shoot to that in the root (S_Na_), K^+^ content in the shoot to that in the root (S_K_), and the Ca^2+^ content in the shoot to that in the root (S_Ca_), and the ratios of the K^+^/Na^+^ ratio in the shoot to that in the root (S_K, Na_) and the Ca^2+^/Na^+^ ratio in the shoot to that in the root (S_Ca, Na_) are shown in Appendix A. It can be seen that the correlations between S_K, Na_, S_Ca, Na_, and the indicators are not significant, or the correlation coefficients are low. In the three ratios of the shoot ion content to the root ion content, S_Na_ showed a higher correlation with the experimental indicators, followed by S_K_, while the correlations between S_Ca_ and the indicators were not significant, or the correlation coefficients were very low (Table 1, Table 2 and Table 3).

Take the linear relationships between the ion content and the NaCl concentrations for the highly salt-tolerant lines, the salt-tolerant grade with 200 mmol NaCl treatment, and the total fresh weight of the *B. napus* lines under different concentrations of NaCl shown in Figure 3 as an example. Both the Ca^2+^ and K^+^ contents in the roots and shoots gradually decreased with increasing NaCl concentrations, while the Na^+^ contents in the roots and shoots increased rapidly, particularly in the roots (Figure 3a). The higher the salt-tolerance grade of the *B. napus* inbred lines, the higher the Ca^2+^ and K^+^ contents and the lower the Na^+^ content (Figure 3b). Under the different NaCl concentrations, the higher the total fresh weight of the *B. napus* inbred lines (the higher the salt tolerance), the higher the Ca^2+^ and K^+^ contents, and the lower the Na^+^ content (Figure 3c).

## 4. Discussion

In the present study, a total of 20 *B. napus* inbred lines were randomly chosen from the five salt-tolerance categories described by Wu, et al. [46]. These lines were used to uncover the underlying mechanisms of salt tolerance, with an emphasis on its correlation between growth and ion parameters. This information would like to provide a basis for the further elucidation of the molecular mechanisms underlying *B. napus* salt tolerance, and for breeding salt-tolerant cultivars.

The salt tolerance of a plant is related to the ion balance of the plant itself. Under salt stress, the Na^+^ content of the plant increases. It is well known that salt stress inhibits plant growth, mainly due to the competition in transport between Na^+^ and other mineral elements such as K^+^ and Ca^2+^ [51,52,53,54]. This interferes with the plant’s ability to absorb other mineral elements, resulting in a deficiency of those elements, most commonly K^+^. At high Na^+^ concentrations, growth and Ca^2+^ uptake are reduced in sugarcane (*Saccharum* sp.), rice (*Oryza sativa*), and wheat (*Triticum aestivum*) [52,55,56]. Even in the relatively salt-tolerant wild rapeseed (*Brassica campestris*), an increased soil Na^+^ concentration is accompanied by decreased Mg^2+^ and K^+^ uptake [57]. Some *B. napus* germplasms have higher Ca^2+^ and K^+^ contents, which may be related to their salt tolerance, because Ca^2+^ and K^+^ absorption by seedlings is considered to be particularly important in high-NaCl saline environments [58]. Na^+^ accumulation in the shoot tissues is regarded as one of the most useful phenotypes in screening for salinity tolerance [7]. Here, our results indicate that the ionic contents in *B. napus* seedlings were also affected by NaCl stress, and seedlings with poor salt tolerance have higher Na^+^ contents and lower K^+^ and Ca^2+^ contents. These changes in the abundances of the Na^+^, K^+^, and Ca^2+^ ions consequently altered the other related ion parameters (K^+^/Na^+^, Ca^2+^/Na^+^, etc.) of the seedlings. By comparing the ion contents with the growth indicators of the five salt-tolerance grades of *B. napus* inbred lines, we found that the better the plant grows under salt stress (the higher salt tolerance), the higher the Ca^2+^ and K^+^ contents and the lower the Na^+^ content in the plant body (Appendix A). In addition, the K^+^/Na^+^ ratio is regarded as a reliable indicator of salt tolerance for many crops [59,60,61,62], and we showed that its highest value in our experiments is in the *B. napus* inbred lines with a higher salt tolerance (Appendix A).

Many studies have been performed on the salt tolerance of plants [63,64,65,66], but few using so many inbred lines with different salt tolerances have been reported. Previous research has focused more on the analysis of the differences in the final physiological indicators, while here we focus on a correlative analysis of the changes in the physiological indicators in particular ion parameters with the salt tolerance of *B. napus* inbred lines under different NaCl concentrations. To date, evaluations of the salt resistance of crops using the most reliable indicators have not yet reached a clear conclusion; for example, some studies showed that the contents of Na^+^ and Ca^2+^ are more related to the yield of seeds than K^+^ [63], while others suggest that the contents of Na^+^ and K^+^ are more important than Ca^2+^ [67].

Here, we approached this issue by taking into account the differences in the salt tolerance of the different inbred lines a single species and analyzed the relationships between the indicators and salt tolerance under various variables using linear regressions. First, we found that all salt-tolerance grades showed a high correlation of about *R*^2^ = 0.7 with the salt concentration on average (Table 1); therefore, about 30% of the differences detected are related to the experimental materials and the external environment. Second, among the plant physiology parameters, higher NaCl concentrations are more strongly correlated with the ion contents, which is presumed to be because low concentrations of salt are not sufficient to cause significant differences in the ion contents and salt tolerances of different varieties of *B. napus* (Table 2). Third, root length is the growth parameter most strongly correlated with ion contents (Table 3). We speculate that this is related to the function of roots in selectively absorbing and transporting ions; that is, the longer the root, the better the plant grows and the more effectively it takes up and transports ions such as K^+^ and Ca^2+^ to the shoot. For Na^+^, a negative trend was found between root growth and salt tolerance. Using a comprehensive analysis comprising linear regression and significant difference tests, we found that, at the *p* < 0.01 level, the *R*^2^ for the shoot Na^+^ content is 0.5 on average, and the *R*^2^ for the root Na^+^ content is 0.57 on average; the *R*^2^ for the shoot K^+^ content is 0.51 on average, and the *R*^2^ for the root K^+^ content is 0.47 on average; the *R*^2^ for the shoot Ca^2+^ content is 0.5 on average, and the *R*^2^ for the root Ca^2+^ content is 0.48 on average. The correlation coefficients for the Ca^+^/Na^+^ and K^+^/Na^+^ ratios are also higher (about 0.4–0.5), while those for S_Ca, Na_, S_K, Na_, S_Na,_ S_K_, and S_Ca_ are lower (about 0.1–0.3). In conclusion, the root Na^+^ content is the most critical indicator of the salt tolerance of *B. napus*.

The root is the first organ of a plant exposed to a salty environment, and plays a momentous role in salt sensing [68,69] and signal transmission to the aerial tissues [70]. From observing plant growth and analyzing our initial data, we found that under salt stress, *B. napus* shoot growth decreases, root growth is poorer, plant growth reduces, and necrosis occurs. This supports that the root system was directly exposed to high concentrations of salt during the growth process, or because the root is more sensitive and thus more easily damaged than the shoot [71]. Later experiments found that salt (Na^+^) can be transported to the aerial parts of the salt-resistant plants, which may be to better protect the roots [72]. Studies have shown that halophytes are more efficient than non-halophytes in physiological regulation such as ions under salt stress [17,73]. In general, halophytes respond to salt stress signals faster, and are more efficient in maintaining membrane potential and voltage gating, which helps them maintain ion and osmotic homeostasis under stress [73,74]. This may also be the reason for the salt tolerance of salt-tolerant varieties. Specifically, in our experiment, the varieties with strong salt tolerance can transport salt ions to the above-ground part or deal with the absorption of salt ions more efficiently leading to less Na^+^ in the roots, while the varieties with weak salt tolerance have lower control efficiency leading to more Na^+^ in the roots.

Plant roots absorb water and inorganic salts from their environment and transport them upward to their aerial organs, which biosynthesize organics that are delivered to the underground parts of the plants. Excessive amounts of salt ions (mainly Na^+^) can damage plant tissues and negatively affect growth indicators, such as root length and fresh weight [75,76]. The elongation of the main root can thus reflect and be used to characterize plant salt tolerance [29]. At the cellular level, in order to shield the highly sensitive biosynthetic devices from harmful excess Na^+^, the concentration of Na^+^ in the cytoplasm must be kept at a relatively low level that can be tolerated by the intracellular machinery [77,78], especially within the aerial parts of the plants. Ion concentration in cytosol of plant cells under salt stress depends upon the import, transportation and deposition [43], which can be reflected in ion selectivity. Among the correlation coefficients between S_Ca, Na_, S_K, Na_, S_Na_, S_K_, S_Ca_, and the various indicators, S_Na_ also had the highest or higher correlation coefficient (*R*^2^ = 0.90 for salt-sensitive varieties in Table 1, *R*^2^ = 0.50 for *B. napus* inbred lines grown under 150 mmol NaCl stress in Table 2, *R*^2^ = 0.54 for the root length of the *B. napus* inbred lines in Table 3).

The uptake, transportation, deposition, and efflux of Na^+^ in the plant roots determines the Na^+^ content and salt tolerance of a plant [79,80]. There are two main pathways by which Na^+^ enters the root xylem vessels from the soil: the apoplastic pathway and the symplastic pathway. There is mounting evidence to indicate that apoplastic transpiration plays a key role in the movement of Na^+^ into the shoots and the resulting salt tolerance, because the apoplastic barriers (Casparian bands and suberin lamellae) in the exodermis and endodermis of roots can block the bypass flow of Na^+^ and prevent it from entering the shoots [81,82,83,84,85]. In the case of the *B. napus* inbred lines, our results indicate that the root Na^+^ content is highly negatively correlated with their salt tolerance. In addition, the K^+^ and Ca^2+^ contents are positively associated with their salt tolerance. These results suggest that root apoplastic barriers may partly participate in the salt tolerance of *B. napus,* which need to further study.

Membrane-bound translocating proteins facilitate ion transport across plant membranes. Many advances have been made in the study of ion transporters in plant salt tolerance. During exposure to salt stress, the salt overly sensitive (SOS) pathway is known to play a vital role in plants, and the exploration of salt stress perception, signal transduction pathways and regulatory mechanisms are all related to this [20,21,86,87]. The main components of this pathway are a plasma membrane Na^+^/H^+^ antiporter (SOS1), a Ser/Thr protein kinase (SOS2), and a Ca^2+^-binding protein (SOS3) [26,88,89]. The ion regulation ability of a plant determines the ion concentration in its tissues; the ability to balance ions between different tissues and cells is a pivotal mechanism for regulating plant salt tolerance [90,91,92,93]. It has been reported that the high-affinity K^+^ transporters (HKTs) HKT1 and HKT2 are involved in the exclusion of Na^+^ [94], and Zeeshan, et al. [95] have confirmed that they can recover Na^+^ from the xylem vessels and limit the transport of Na^+^ to the shoots. Peng, et al. [91] found that the activities of the Na^+^/H^+^ antiporter and H^+^-ATPase are pivotal factors that determine the ability of the plant to partition ions into its leaves. Salt stress can be related to the regulation of H^+^-ATPase in the plasma membrane of the roots and leaves. H^+^-ATPase, H^+^-PPase, and the Na^+^/H^+^ reverse transport proteins in the tonoplast can accelerate the absorption of K^+^ and Ca^2+^, the release of Na^+^, and the accumulation of Na^+^ in the vacuole, thus promoting the regional distribution of salt [96,97,98,99,100]. Most commonly, by isolating toxic amounts of Na^+^ into the vacuoles through the action of the Na^+^/H^+^ antiporters, the maintenance of cytosolic Na^+^ at a tolerable concentration is achieved in non-halophytes [94,101]. Halophytes can even accumulate ions in vacuoles for osmotic adjustment to cope with saline habitats [17,102]. In the present research, significant differences were detected in the ion contents and selective absorption and transport in the *B. napus* inbred lines with different salt tolerances, which were somewhat correlated with their salt tolerance (Appendix A). This suggests that membrane-bound translocating proteins in the roots, such as SOS1, Na^+^/H^+^ exchanger 1 (NHX1), HKT1, HKT2, and H^+^-ATPase also play a part in the uptake of Na^+^, K^+^, and Ca^2+^ and their transport in *B. napus* under salt stress, which requires further exploration.

## 5. Conclusions

Under NaCl stress, the growth of rape seedlings was inhibited. Salt stress leads to an increase in Na^+^ content and decrease of K^+^, Ca^2+^ content of the seedlings. We calculated the correlation coefficients of the growth parameters and ion parameters under different concentrations of NaCl (Appendix A). We found that the salt tolerance of the *B. napus* has a linear relationship with salt concentrations. In addition, our results indicate that the physiological indicators of *B. napus* seedlings, such as the ion content and ion selective transport, changed slightly at low salt concentrations, while only small effects were detected for the inhibition of plant growth and no significant difference was found among different salt-tolerance categories. The physiological indicators changed more significantly with the increase of salt concentration, as did the significant differences among different salt-tolerance categories were greater. Our analysis of the correlation coefficients indicated that the Na^+^ content is the pivotal factor, particularly the root Na^+^ content, for the salt tolerance of *B. napus*; the higher the salt tolerance, the lower the root Na^+^ content. The Ca^2+^/Na^+^ and K^+^/Na^+^ ratios of the roots were also highly correlated with salt tolerance in *B. napus*.

## Figures and Tables

**Figure 1 plants-11-00906-f001:**
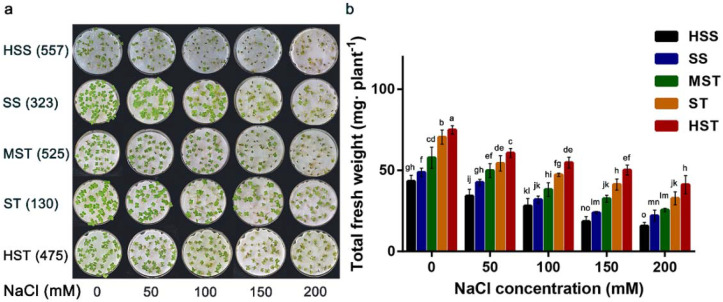
Seedling growth phenotypes of five *B. napus* lines at five salt tolerance levels. (**a**) Phenotypes of the seedlings with high salt sensitivity (HSS, 557); salt sensitivity (SS, 323); moderate salt tolerance (MST, 525); salt tolerance (ST, 130); and high salt tolerance (HST, 475) at different NaCl concentrations. (**b**) Average total fresh weight of the *B. napus* lines when grown at different NaCl concentrations. Error bars indicate standard deviation (*n* = 4). Statistical significance was determined using a one-way ANOVA and Duncan’s multiple range test. Significant differences at *p* < 0.05 are represented by different letters above the bars. 557, 323, 525, 130, and 475, these numbers represent the number of the *B. napus* lines.

**Figure 2 plants-11-00906-f002:**
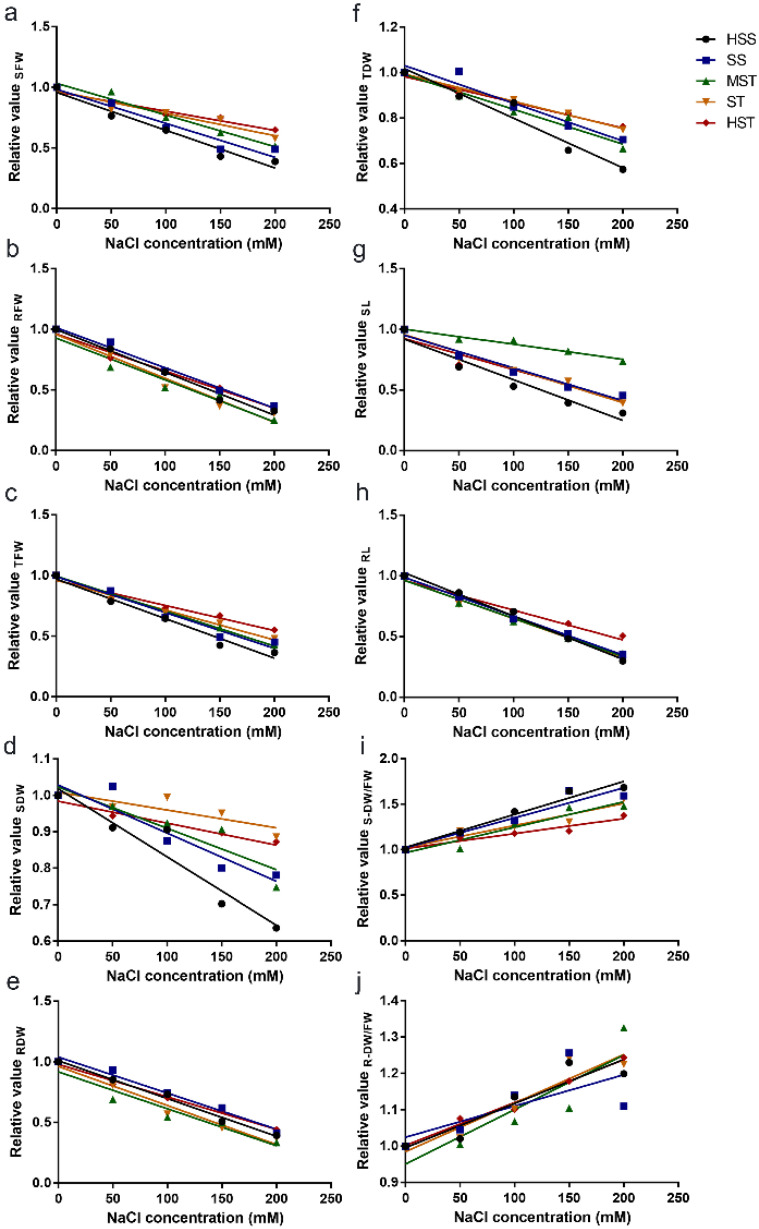
Linear relationship between the NaCl concentration and the relative values of the various growth indicators. The indicators measured were the shoot fresh weight (SFW; (**a**)), root fresh weight (RFW; (**b**)), total fresh weight (TFW; (**c**)), shoot dry weight (SDW; (**d**)), root dry weight (RDW; (**e**)), total dry weight (TDW; (**f**)), shoot length (SL; (**g**)), root length (RL; (**h**)), ratio of dry weight to fresh weight of the shoots (S-DW/FW; (**i**)), and ratio of dry weight to fresh weight of the roots (R-DW/FW; (**j**)). Each point represents the mean of four replicates. HSS, high salt sensitivity; SS, salt sensitivity; MST, moderate salt tolerance; ST, salt tolerance; HST, high salt tolerance.

**Figure 3 plants-11-00906-f003:**
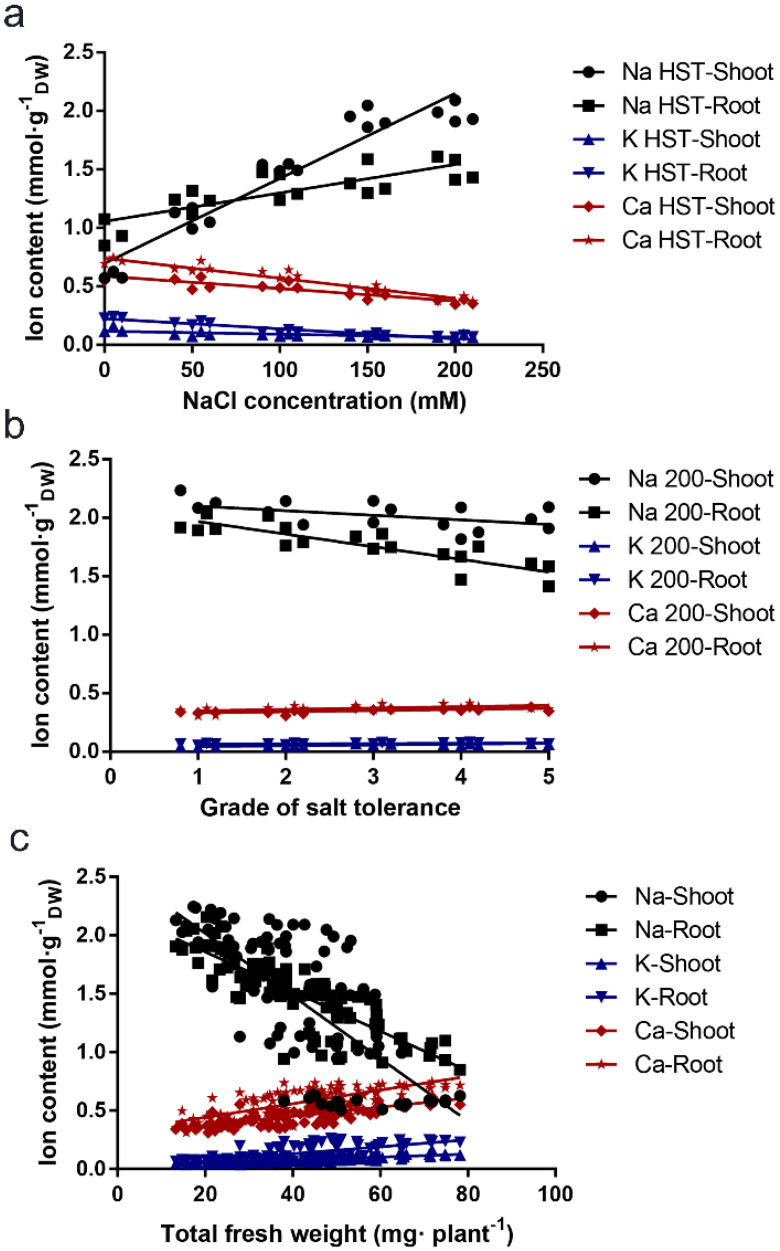
Correlations between the ion content and the NaCl concentration, the grade of salt tolerance, or the growth indicator in *B. napus* seedlings. (**a**) Correlations between the different concentrations of NaCl (mmol) and Ca^2+^, K^+^, and Na^+^ contents in *Brassica napus* seedlings with high salt tolerance (HST). (**b**) Correlations between the salt tolerance category and the Ca^2+^, K^+^, and Na^+^ contents in *B. napus* seedlings grown at 200 mmol NaCl. (**c**) Correlations between the growth indicator total fresh weight (TFW) and the Ca^2+^, K^+^, and Na^+^ contents in *B. napus* seedlings grown under different NaCl concentrations. Salt tolerance category: 1, HSS, high salt sensitivity; 2, SS, salt sensitivity; 3, MST, moderate salt tolerance; 4, ST, salt tolerance; 5, HST, high salt tolerance.

**Table 1 plants-11-00906-t001:** Correlation coefficients between the shoot and root ion contents, absorption, and transport, and the NaCl concentrations in the *B. napus* inbred lines of five salt-tolerance categories.

	Na		K		Ca	
	Shoot	Root	Shoot	Root	Shoot	Root
HSS	0.94 **	0.85 **	0.71 **	0.89 **	0.90 **	0.95 **
SS	0.91 **	0.80 **	0.75 **	0.88 **	0.88 **	0.93 **
MST	0.95 **	0.85 **	0.72 **	0.89 **	0.90 **	0.93 **
ST	0.92 **	0.79 **	0.76 **	0.84**	0.89 **	0.90 **
HST	0.94 **	0.71 **	0.56 **	0.91 **	0.85 **	0.93 **
	**K/Na**		**Ca/Na**		**S_K, Na_**	**S_Ca, Na_**
	Shoot	Root	Shoot	Root		
HSS	0.70 **	0.82 **	0.78 **	0.90 **	0.10	0.57 **
SS	0.68 **	0.82 **	0.76 **	0.90 **	0.04	0.57 **
MST	0.74 **	0.84 **	0.81 **	0.90 **	2.51 × 10^−4^	0.65 **
ST	0.72 **	0.81 **	0.81 **	0.94 **	0.20	0.65 **
HST	0.67 **	0.86 **	0.80 **	0.92 **	0.02	0.60 **
	**S_Na_**	**S_K_**	**S_Ca_**			

HSS	0.85 **	0.49 **	0.30 *			
SS	0.86 **	0.70 **	0.30 *			
MST	0.88 **	0.71 **	0.64 **			
ST	0.90 **	0.60 **	0.31 *			
HST	0.80 **	0.69 **	0.41 **			

*, ** = significant at 0.05 and 0.01 probability levels, respectively. HSS, high salt sensitivity; SS, salt sensitivity; MST, moderate salt tolerance; ST, salt tolerance; and HST, high salt tolerance.

**Table 2 plants-11-00906-t002:** Correlation coefficients between the shoot and root ion contents, absorption, and transport, and different salt-tolerance grades of *B. napus* under various NaCl concentrations.

	Na		K		Ca	
	Shoot	Root	Shoot	Root	Shoot	Root
0	6.04 × 10^−6^	0.06	0.34 **	0.02	0.01	0.08
50	0.09	0.67 **	0.32 **	0.02	0.08	0.15
100	0.25 *	0.61 **	0.56 **	0.10	0.35 **	0.20 *
150	0.19	0.74 **	0.60 **	0.18	0.35 **	0.27 *
200	0.25 *	0.75 **	0.65 **	0.26 *	0.44 **	0.33 **
	**K/Na**		**Ca/Na**		**S_K, Na_**	**S_Ca, Na_**
	Shoot	Root	Shoot	Root		
0	0.30 *	0.10	1.54 × 10^−3^	0.10	0.11	0.07
50	0.31 *	0.53 **	0.15	0.69 **	0.06	0.13
100	0.59 **	0.36 **	0.44 **	0.77 **	0.13	0.02
150	0.65 **	0.69 **	0.41 **	0.78 **	0.02	0.54 **
200	0.74 **	0.73 **	0.49 **	0.72 **	0.19	0.34 *
	**S_Na_**	**S_K_**	**S_Ca_**			

0	0.07	0.26 *	4.28 × 10^−3^			
50	0.24 *	0.40 **	0.01			
100	0.32 **	0.35 **	0.11			
150	0.50 **	0.39 **	0.01			
200	0.45 **	0.44 **	2.93 × 10^−3^			

*, ** = significant at 0.05 and 0.01 probability levels, respectively.

**Table 3 plants-11-00906-t003:** Correlation coefficients between the growth indexes and the ion content parameters of all *B. napus* varieties under all treatments.

	Na		K		Ca	
	Shoot	Root	Shoot	Root	Shoot	Root
TFW	0.53 **	0.71 **	0.64 **	0.49 **	0.55 **	0.50 **
TDW	0.38 **	0.48 **	0.44 **	0.31 **	0.34 **	0.38 **
SFW	0.38 **	0.58 **	0.55 **	0.34 **	0.40 **	0.36 **
RFW	0.69 **	0.73 **	0.63 **	0.68 **	0.68 **	0.64 **
SDW	0.19 **	0.27 **	0.25 **	0.14 **	0.15 **	0.20 **
RDW	0.65 **	0.72 **	0.62 **	0.63 **	0.67 **	0.62 **
SL	0.57 **	0.46 **	0.55 **	0.54 **	0.58 **	0.49 **
RL	0.76 **	0.70 **	0.63 **	0.75 **	0.73 **	0.75 **
S-DW/FW	0.41 **	0.66 **	0.55 **	0.39 **	0.51 **	0.45 **
R-DW/FW	0.44 **	0.36 **	0.22 **	0.42 **	0.36 **	0.41 **
	**K/Na**		**Ca/Na**		**S_K, Na_**	**S** ** _Ca, Na_ **
	Shoot	Root	Shoot	Root		
TFW	0.50 **	0.54 **	0.48 **	0.63 **	0.11 **	0.24 **
TDW	0.31 **	0.34 **	0.30 **	0.42 **	0.11 **	0.14 **
SFW	0.36 **	0.38 **	0.33 **	0.48 **	0.11 **	0.14 **
RFW	0.62 **	0.71 **	0.64 **	0.77 **	0.07 **	0.41 **
SDW	0.15 **	0.15 **	0.13 **	0.21 **	0.09 **	0.05 *
RDW	0.58 **	0.68 **	0.60 **	0.76 **	0.07 **	0.34 **
SL	0.57 **	0.54 **	0.58 **	0.55 **	0.16 **	0.49 **
RL	0.64 **	0.72 **	0.68 **	0.78 **	0.07 **	0.47 **
S-DW/FW	0.36 **	0.42 **	0.36 **	0.53 **	0.06 *	0.14 **
R-DW/FW	0.28 **	0.36 **	0.34 **	0.38 **	1.81 × 10^−3^	0.27 **
	**S** ** _Na_ **	**S** ** _K_ **	**S** ** _Ca_ **			

TFW	0.24 **	0.08 **	0.11 **			
TDW	0.17 **	0.04	0.14 **			
SFW	0.13 **	0.02	0.08 **			
RFW	0.44 **	0.25 **	0.14 **			
SDW	0.07 **	2.19 × 10^−3^	0.10 **			
RDW	0.38 **	0.20 **	0.13 **			
SL	0.46 **	0.18 **	0.09 **			
RL	0.54 **	0.32 **	0.23 **			
S-DW/FW	0.13 **	0.04 *	0.10 **			
R-DW/FW	0.33 **	0.31 **	0.14 **			

*, ** = significant at 0.05 and 0.01 probability levels, respectively. TFW, total fresh weight; TDW, total dry weight; SFW, fresh weight of shoots; RFW, fresh weight of roots; SDW, dry weight of shoots; RDW, dry weight of roots; SL, shoot length; RL, root length; S-DW/FW, dry weight to fresh weight of shoots ratio; R-DW/FW, dry weight to fresh weight of roots ratio.

## Data Availability

Data is contained within the article.

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
