# Peer review of "Root Na+ Content Negatively Correlated to Salt Tolerance Determines the Salt Tolerance of Brassica napus L. Inbred Seedlings"

_plants, 2022, doi:10.3390/plants11070906_

Round 1
Reviewer 1 Report
It is necessary to state that these results are obtained because only sodium chloride solution is used in the experiments. In saline soils, however, it is not at all certain that there is only the presence of saline sodium. This can only happen in some alkaline soils
Ιn the introduction a distinction needs to be made between saline and alkaline soils because in alkaline soils the presence of sodium is important
l. 96-97. It is more appropriate to put the thanks at the end of the article
l.260. Give clarifications about the symbolisms
Author Response
1) Comments:
It is necessary to state that these results are obtained because only sodium chloride solution is used in the experiments. In saline soils, however, it is not at all certain that there is only the presence of saline sodium. This can only happen in some alkaline soils
Response: Thanks for the review for us. Yes, saline-alkali land is a general term for various saline soils and alkaline soils. NaCl is only one of the most common components in saline soils, and only NaCl was involved in our experiments. We have indicated in the Abstract, Introduction, and Materials and Methods that the salt used in the experiments is NaCl, details were provided in lines 15-16, 89-92, 108-110.
2) Comments:
Ιn the introduction a distinction needs to be made between saline and alkaline soils because in alkaline soils the presence of sodium is important
Response: Thanks for the review for us. As mentioned in the question above, saline-alkali land is a general term for various saline soils and alkaline soils. Sodium salts are the most common in saline-alkali land. The presence of sodium in alkaline soils is indeed important, but we are not going to study the difference between saline soils and alkaline soils in detail, but to study some related problems of their salt tolerance for salts in soils. And the salt used in our experiments was NaCl. We have made a simple explanation in the introduction, which was provided in lines 29-30.
3) Comments:
- 96-97. It is more appropriate to put the thanks at the end of the article
Response: Thanks for the review for us. It has been pointed out in the acknowledgments section of the article, which was provided in lines 449-451.
4) Comments:
l.260. Give clarifications about the symbolisms
Response: Thanks for the review for us. These are some ion indicators mentioned in past reports, and the specific meanings were provided in lines 252-257. The description of ionic indicators has been mentioned in the introduction and discussion section. These ion indicators were also shown several times in the chart data in the text.

Reviewer 2 Report
The presented manuscript is a comprehensive statistical study of the influence of salinity on the growth parameters and the content of selected ions in the underground and above-ground parts of Brassica napus L. plants and their correlation with salinity tolerance. 20 rape inbred lines belonging to 5 salinity tolerance categories were selected for the analysis.
The work has significant application value and indicates specific parameters (Na + ion content in roots) that are related to salinity tolerance. Although the work is interesting, I have a few comments:
figures 1a, 2 and 3 are illegible. I suggest enlarging them and improving the quality (especially photos of seedlings).
Figures 2 and 3 lack a statistical analysis showing whether the observed trends are in fact different from each other.
In the discussion in paragraph 392-416, there is no information about HKT1 and HKT2 proteins, which are crucial in the transport and uptake of Na + ions.
The authors attached 3 files with supplements, while only two of them are described in the text and in the supplementary material section (433-436).
Moreover, the authors made some linguistic mistakes or awkwardness, for example: line 47 - "germination stop" - rather inhibition; lines 49-51 - incomprehensible sentence; line 114 - "growth house"; line 126 - "were killed"; line 428-429 - "correlative analysis of correlation coefficients".
Moreover, in the descriptions of the Tables there should be explanations of the abbreviations used, and in the description of Figure 3, the degrees of salinity tolerance (1-5) should be described (instead the authors have explained the abbreviations that were not used here). In addition, the description of table 1 "salt tolerant lines" is incorrect, because the table applies to both tolerant and sensitive lines.
Author Response
1) Comments:
The presented manuscript is a comprehensive statistical study of the influence of salinity on the growth parameters and the content of selected ions in the underground and above-ground parts of Brassica napus L. plants and their correlation with salinity tolerance. 20 rape inbred lines belonging to 5 salinity tolerance categories were selected for the analysis.
The work has significant application value and indicates specific parameters (Na + ion content in roots) that are related to salinity tolerance. Although the work is interesting, I have a few comments:
figures 1a, 2 and 3 are illegible. I suggest enlarging them and improving the quality (especially photos of seedlings).
Response: Thanks for the review and suggestion for us. We made the figures bigger and clearer. And we would resubmit 3 original figures that are clearer.
2) Comments:
Figures 2 and 3 lack a statistical analysis showing whether the observed trends are in fact different from each other.
Response: Thanks for the review for us.
Figure 2 is the analysis of the linear relationship between the salt concentration and the relative value of the growth indexes under salt stress. Details can be found in Supplementary file 1. The analysis results of Figure 2 were provided in lines 197-206.
Figure 3 is an example of the linear relationships between the ion content and the NaCl concentrations for the highly salt-tolerant lines, the salt-tolerant grade with 200 mmol NaCl treatment, and the total fresh weight of the B. napus lines under different concentrations of NaCl. The totality of data is too much, and we show it in the Supplementary files as well as in the Non-published Data. The analysis of what is shown in Figure 3 in the text is provided in lines 263-272.
In the Results section, a preliminary analysis and summary of the conclusions that can be drawn from the data contained in the figures and tables are presented. In addition to the intuitive results, the further analysis and discussion of the combined figures and tables are described in detail in the Discussion section.
3) Comments:
In the discussion in paragraph 392-416, there is no information about HKT1 and HKT2 proteins, which are crucial in the transport and uptake of Na + ions.
Response: Thanks for the suggestion for us. We have supplemented the information on HKT1 and HKT2 proteins. The details were provided in lines 398-401.
4) Comments:
The authors attached 3 files with supplements, while only two of them are described in the text and in the supplementary material section (433-436).
Response: Thanks for the review for us. There are only 2 Supplementary files, while the Non-published data has 3 files. Therefore, only Supplementary files 1 and 2 are listed in the Supplementary Materials section.
5) Comments:
Moreover, the authors made some linguistic mistakes or awkwardness, for example: line 47 - "germination stop" - rather inhibition; lines 49-51 - incomprehensible sentence; line 114 - "growth house"; line 126 - "were killed"; line 428-429 - "correlative analysis of correlation coefficients".
Response: Thanks for the review for us. We made changes to remove ambiguity.
Example 1: Here is to show that external stress can inhibit the germination process of seeds, or even kill the seeds, so that the seeds can no longer germinate. The details of the modification were provided in line 48.
Example 2: The modification to this sentence was provided in lines 50-52.
Example 3: We have made a change, which was provided in line 113.
Example 4: This is a step in the drying process of plant materials. We modified this in line 125.
Example 5: We modified this in line 428.
6) Comments:
Moreover, in the descriptions of the Tables there should be explanations of the abbreviations used, and in the description of Figure 3, the degrees of salinity tolerance (1-5) should be described (instead the authors have explained the abbreviations that were not used here). In addition, the description of table 1 "salt tolerant lines" is incorrect, because the table applies to both tolerant and sensitive lines.
Response: Thanks for the review for us. The description in Figure 3 has been modified, which was provided in lines 280-281. And in the descriptions of the Tables, we added the explanations of the abbreviations used.

This manuscript is a resubmission of an earlier submission. The following is a list of the peer review reports and author responses from that submission.
Round 1
Reviewer 1 Report
The subject of the article is interesting but the manuscript is far from the standards of a scientific paper.
Typical examples are the lack of presentation of the aims and the object of the work but also the wrong structure of the article. The manuscript first presents the results of the experiments and then the materials and methods section. Also the section of conclusions is insufficient (5 lines with generalities). In the results chapter they make a simple description of the results without delving into the subject. In the introduction they must document the gap that their work is coming to fill.
Reviewer 2 Report
Dear authors,
This is an interesting and well written manuscript. I really enjoyed reading this research and the dataset is good.
- All key elements are present.
- The title clearly describes the article.
- The abstract content clearly reflects the entire content of the article.
- In the introduction paragraph the authors clearly estate the problem investigated. Also, the purpose of the study is specified.
- The authors accurately explain the field experiments.
- Data are well presented
Reviewer 3 Report
The authors present the characterization of the 10- days growth of five genotypes of Brassica napus, in response to increased levels of NaCl concentration (5 levels of concentration).
First of all, the manuscript is pleasant to read: the work is well introduced and a nice discussion is proposed. Part of the work is based on previously acquired and published data (Wu et al., 2019). The ambition of this manuscript was to complete the analysis with ionic data in order to “uncover the underlying mechanisms of salt tolerance”. On this point, the manuscript does not fulfill its ambitions, because it remains very descriptive and does not present functional data that could explain the associated mechanisms.
If the ambition was to find a new indicator of salt tolerance (that is Root Na+ content), then we could say that the manuscript meets its objectives, but only partly as another experiment should “validate” this indicator in an independent manner.
If the ambition remains to uncover the underlying mechanisms, then other physiological data should be given on the same genotypes, in order to avoid speculation.
Concerning the choice of genotypes, it was very smart to choose genotypes from contrasting tolerance groups, but why did you choose these genotypes randomly? By doing so, the genotypes with contrasting tolerance also show a proportional growth gradient in the control condition: the most tolerant are the smallest genotypes. Choosing genotypes with the same growth under control conditions would help to have a stronger demonstration.
Here are other minor comments :
In the whole manuscript, you need to define abbreviations before their first use, and you need to redefine them in the legend of each table.
Introduction
- Line 32 : what percentage of arable land correspond to the surfaces of salinized soil?
- Line 63 : replace “;” by “.”
- Line 63-64 : Give more information about what is shoot-ion independent tolerance
Results
- Line 122: “whereas those of the other indicators were…” : precise which indicators you are talking about
- Line 121 : define the abbreviations
- Figure 2 : the trait that corresponds to each sub-figure should be more visible. Could you add the statistical analysis on this figure rather than in a table ?
- There are too many tables that look the same and that induce confusion. Could you propose a more synthetic way to present these data ? Correlation matrix for example, with hierarchical clustering of the traits?
Discussion:
- Figure 4 is not informative and can be removed
- Could you discuss about the fact that your indicator is located in the roots ? Is it as easy to use as an indicator located in the aerial part for breeding purpose for example?
Material and Methods:
- Lines 414-415 : The unit should be added to gain in precision. Indeed, content is a term that can be used for a quantity or a concentration. By adding the unit of measure, then the ambiguity is removed.
- Line 439 : “specific tools including mathematical models” : you need to be more precise and to describe this model.